# Nanostructured Modifications of Titanium Surfaces Improve Vascular Regenerative Properties of Exosomes Derived from Mesenchymal Stem Cells: Preliminary In Vitro Results

**DOI:** 10.3390/nano11123452

**Published:** 2021-12-20

**Authors:** Chiara Gardin, Letizia Ferroni, Yaşar Kemal Erdoğan, Federica Zanotti, Francesco De Francesco, Martina Trentini, Giulia Brunello, Batur Ercan, Barbara Zavan

**Affiliations:** 1Maria Cecilia Hospital, GVM Care & Research, Cotignola, 48033 Ravenna, Italy; cgardin@gvmnet.it (C.G.); lferroni@gvmnet.it (L.F.); 2Biomedical Engineering Program, Middle East Technical University, Ankara 06800, Turkey; yasarer@metu.edu.tr (Y.K.E.); btrercan@gmail.com (B.E.); 3Department of Biomedical Engineering, Isparta University of Applied Science, Isparta 32260, Turkey; 4Department of Translational Medicine, University of Ferrara, 44121 Ferrara, Italy; zntfdc@unife.it (F.Z.); tntmrt@unife.it (M.T.); 5Department of Plastic and Reconstructive Surgery-Hand Surgery Unit, Azienda ‘Ospedali Riuniti’, 60126 Ancona, Italy; fran.defr@libero.it; 6Department of Neurosciences, Dentistry Section, University of Padova, 35128 Padova, Italy; giulia-bru@libero.it; 7Department of Oral Surgery, University Clinic Düsseldorf, 40225 Dusseldorf, Germany; 8Department of Metallurgical and Materials Engineering, Middle East Technical University, Ankara 06800, Turkey; 9BIOMATEN, METU Center of Excellence in Biomaterials and Tissue Engineering, Ankara 06800, Turkey

**Keywords:** titanium nanotubes, endothelialization, mesenchymal stem cells, endothelial cells, angiogenesis, cardiovascular metal stents

## Abstract

(1) Background: Implantation of metal-based scaffolds is a common procedure for treating several diseases. However, the success of the long-term application is limited by an insufficient endothelialization of the material surface. Nanostructured modifications of metal scaffolds represent a promising approach to faster biomaterial osteointegration through increasing of endothelial commitment of the mesenchymal stem cells (MSC). (2) Methods: Three different nanotubular Ti surfaces (TNs manufactured by electrochemical anodization with diameters of 25, 80, or 140 nm) were seeded with human MSCs (hMSCs) and their exosomes were isolated and tested with human umbilical vein endothelial cells (HUVECs) to assess whether TNs can influence the secretory functions of hMSCs and whether these in turn affect endothelial and osteogenic cell activities in vitro. (3) Results: The hMSCs adhered on all TNs and significantly expressed angiogenic-related factors after 7 days of culture when compared to untreated Ti substrates. Nanomodifications of Ti surfaces significantly improved the release of hMSCs exosomes, having dimensions below 100 nm and expressing CD63 and CD81 surface markers. These hMSC-derived exosomes were efficiently internalized by HUVECs, promoting their migration and differentiation. In addition, they selectively released a panel of miRNAs directly or indirectly related to angiogenesis. (4) Conclusions: Preconditioning of hMSCs on TNs induced elevated exosomes secretion that stimulated in vitro endothelial and cell activity, which might improve in vivo angiogenesis, supporting faster scaffold integration.

## 1. Introduction

Regenerative medicine is the branch of translational medicine aimed at managing and coordinating the phases of regeneration in order to obtain the best functional and aesthetic results for the patient. To this end, different specializations are involved, including biotechnology chemists and engineers who develop solutions by working in coordination, mainly represented by biomaterials, which best resolve the healing process of the wound. Biomaterials must interact with cells in order to mediate the processes of inflammation, cell adhesion, angiogenesis, and production of the extracellular matrix. In all of these processes, the most important key factor affecting the success of scaffold integration is the endothelialization, a process that is still poorly understood [1].

Several biomaterials have been developed, with the most applied being titanium (Ti). Titanium-based scaffolds are routinely used in both orthopedics and the cardiovascular field. Ti alloys are some of the most commonly used materials for clinical implantation due to their superior mechanical strength, corrosion resistance, and biocompatibility [2,3]. More importantly, the surfaces of Ti alloys can be easily modified to have micro- and nanolevels of organization to mimic the natural structure of healthy vessel walls, making them more cytocompatible [4]. Over the years, many surface modifications have been explored for Ti-based alloys, including plasma spraying, grit blasting, acid etching, sand blasting, and the application of bioactive coatings [5,6,7,8,9]. However, these strategies typically generate surface modifications only at the microscale. In contrast, nanostructured surfaces, possessing a higher surface-to-volume ratio, have greater biological plasticity compared to structures containing modifications at the microscale. A simple and inexpensive way to produce nanostructured features on Ti alloys is through electro-chemical anodization, which allows the formation of self-organized, highly ordered, and well-aligned nanotubular structures on the metal surfaces [10,11]. Depending on the electrochemical parameter utilized during the anodization process, different nanofeature morphologies and dimensions can be obtained. Moreover, fabricating nanofeatured surfaces on bare metal implants significantly affects the cellular functions, including adhesion, proliferation, and migration, and stimulates several molecular and cellular events at the tissue–implant interface [12,13,14]. For instance, the fabrication of oxide-based nanotubular structures on Ti alloys enhanced endothelial cell proliferation and increased the activation of angiogenic factors, while preventing the attachment and proliferation of smooth muscle cells on nanostructure surfaces [11,15]. In fact, enhanced and rapid re-endothelialization of bare metal stents via the fabrication of nanophase surface structures was proposed to decrease stent restenosis and platelet adhesion to improve their safety, longevity, and biocompatibility [16,17].

Human mesenchymal stem cells (hMSCs) isolated from different sources exhibited great potential in advanced cell therapies involving Ti-based materials. It was observed that hMSCs secrete vasculogenic growth factors, which promote migration, proliferation, and differentiation of endothelial cells [18,19]. Besides the long-time notion that soluble factors are an important part of the cellular secretome, it now appears that hMSC-derived exosomes, by partially mimicking the properties and function of their parent cells, are mainly responsible for the observed therapeutic effects [20,21]. Exosomes are membrane-bound extracellular vesicles, typically sized in the 30–150 nm range, which are released by all cell types into the extracellular milieu upon the fusion of multivesicular bodies with the plasma membrane [22]. The cargo of the exosomes can include proteins, lipids, mRNA/miRNA, and DNA, and it is highly dependent on the cell type, origin, and microenvironment [23,24]. The hMSC-secreted exosomes seem to be implicated in the angiogenic process in a similar way to their parental cells, since the use of conditioned medium depleted of exosomes impaired the angiogenic hMSC response [25].

The aim of the present work was to evaluate whether nanostructured modifications of cardiovascular stent materials influence secretory functions of hMSCs, and whether these secretions, in turn, affect endothelial cells activities in vitro. To address these questions, 25 nm, 80 nm, and 140 nm diameter nanotubular titanium oxide surfaces (TNs) were fabricated through electrochemical anodization and compared to untreated Ti alloy surfaces. We explored the possibility that exosomes secreted by hMSCs grown on the different TN substrates may act as biomimetic tools for modulating the biological properties of human umbilical vein endothelial cells (HUVECs) in vitro. Exosomes were isolated from the conditioned media of hMSCs seeded on the Ti alloy surfaces, then characterized by transmission electron microscopy (TEM) and flow cytometry. The effects of the different exo-hMSCs preparations on the biological activities of endothelial cells was assessed in terms of HUVECs migration and expression of endothelial cell-specific markers at 48 h post incubation. In addition, the miRNA contents of hMSC-derived exosomes were profiled to elucidate functional mechanisms and angiogenic effects on endothelial cells.

## 2. Materials and Methods

### 2.1. Fabrication of Nanostructured Surfaces

Ti6Al7Nb foil (ISO 5832-11) was cut into 1 × 1 mm^2^ sized samples, followed by sonicating in acetone, 70% ethanol, and distilled water each for 10 min. Prior to anodization, the Ti6Al7Nb samples were etched with 1.5% HF/1.5% HNO_3_ solution for 1 min to remove the naturally occurring oxide layer. A two-electrode electrochemical system was used to anodize the samples, whereby a platinum mesh was used as the cathode and the cleaned sample was the anode. Anodization process took place inside a 0.75% HF + 7.5% H_3_PO_4_ electrolyte solution. The potential was applied at 10 and 20 V for 10 min and 30 V for 5 min [26]. At the end of the process, the anodized samples were rinsed using deionized water and dried at room temperature.

### 2.2. Characterization of Nanostructured Surfaces

Surface morphologies were investigated using a scanning electron microscope (SEM, FEI, Nova Nano 430, NY, USA). Here, 20 kV accelerating voltage was used to image the surfaces and the micrographs were captured using a secondary electrons detector. Nanotubular size measurements were completed using ImageJ software (NIH) by analyzing the diameters of at least 60 different nanotubular features in triplicate for each sample group. The surface nanoscale roughness measurements were performed with an atomic force microscope (AFM, Veeco, Multi-mode V, Santa Barbara, CA, USA). Each surface was investigated in tapping mode using a silicon AFM tip with a 10 nm radius. Here, 0.9 × 0.9 μm^2^ areas were scanned at a scan rate of 1 Hz. The root-mean-square roughness values and surface areas of samples were calculated by Image Plus software. Additionally, the micron-phase roughness values of the samples were measured using a roughness measuring instrument (MarSurf PS 10) from at least three different locations. The chemical composition of the surface layer was examined via X-ray photoelectron spectroscopy (XPS; PHI, 5000 Versa Probe, MIAMI, USA) using a monochromatic Al Kα X-ray source. The surfaces of the samples were sputtered using an Ar+ ion beam at 2 keV for 1 min. Scans for Ti, Nb, Al, and O orbitals were collected and the C 1s peak at 284.8 eV was used as the reference. Deconvolution analysis was performed using XPS Peak41 software. Water contact angles were measured using a goniometer (EasyDrop, KRUSS GmbH, Hamburg, Germany). Here, 4 μL distilled water was dropped onto each sample and the contact angles between the surface and water droplet were measured. These analyses were repeated in triplicate with three different measurements for each sample.

### 2.3. Culture of MSC onto TNs

The hMSCs derived from bone marrow (ATCC, Manassas, VA, USA) were cultured in Dulbecco’s modified Eagle’s medium high-glucose (DMEM HG, EuroClone, Milan, Italy) completed with 10% fetal bovine serum (FBS, EuroClone) and 1% penicillin/streptomycin (P/S, EuroClone), then maintained at 37 °C and 5% CO_2_ with culture medium changes twice a week. At confluence, cells were harvested by trypsin treatment (tryp-sin/EDTA, EuroClone) and counted using a Countess^®^ II Automated Cell Counter (Thermo Fisher Scientific, Waltham, MA, USA). Cells at passage 4 were then seeded onto treated and untreated Ti alloy surfaces at a density of 5 × 10^4^ cells/surface in 12-well plastic tissue culture plates (Corning, VWR International, Milan, Italy). The cells were cultured in DMEM HG completed with 10% exosome-depleted FBS (Thermo Fisher Scientific) at 37 °C with 5% CO_2_ for 7 days, with medium changes every 3 days.

### 2.4. SEM

The hMSCs grown on the different TN surfaces for 7 days were fixed in 2.5% glutaraldehyde prepared in 0.1 M cacodylate buffer for 1 h, then progressively dehydrated in ethanol. All micrographs were obtained using a JSM JEOL 6490 SEM microscope (JEOL, Tokyo, Japan). The SEM analysis was performed at Centro di Analisi e Servizi per la Certificazione (CEASC, University of Padova, Padova, Italy).

### 2.5. RNA Isolation from hMSCs, First-Strand cDNA Synthesis, and Real-Time PCR

Total RNA was isolated with the total RNA purification Plus kit (Norgen Biotek, Thorold, ON, Canada) from hMSCs seeded onto TNs for 7 days. The RNA quality and concentrations of the samples were measured with the NanoDrop™ One (Thermo Fisher Scientific). For the first-strand cDNA synthesis, 800 ng of RNA was reverse-transcribed using the SensiFAST™ cDNA Synthesis Kit (Bioline GmbH, Luckenwalde, Germany) in a LifePro Thermal Cycler (Bioer Technology, Hangzhou, China). Real-time PCR was then performed to analyze the expression levels of genes coding for angiogenic-related factors (Table 1). Real-time PCR was carried out using the designed primers at a concentration of 400 nM and SensiFAST™ SYBR No-ROX mix (Bioline GmbH) on a Rotor-Gene Q (Qiagen, Hilden, Germany). Data analysis was performed using the 2^ΔΔCt^ method [27] using transferrin receptor (TFRC) as the internal reference. Data are presented as mean fold changes with respect to control samples (hMSCs seeded onto tissue culture polystyrene (TCP) for 7 days).

### 2.6. Exosomes Isolation from Cells Grown onto TNs

Exo-hMSCs were isolated from cells grown onto the different TNs for 7 days using the Cell Culture Media Exosome Purification Kit (Norgen Biotek), according to the manufacturer’s instructions. Exosomes were then used for downstream analysis or stored at –80 °C until further use.

### 2.7. Characterization of exo-hMSCs

For TEM, a drop (20 μL) of exosome suspension was applied and allowed to adsorb for 2 min to 300 mesh Formvar carbon-coated copper grids. Excess liquid was removed with filter paper (Whatman, Maidstone, UK), then samples were negatively stained with 1% uranyl acetate for 2 min. The grids were observed with a FEI Tecnai G2 transmission electron microscope operating at 100 kV (Department of Biology, University of Padua, Italy), and the images captured with a Veleta (Olympus Soft Imaging System, Münster, Germany) digital camera.

For flow cytometry, CD81-positive exosomes were isolated with Exosome-Human CD81 Flow Detection (from cell culture) (Thermo Fisher Scientific), according to the manufacturer’s instructions. Briefly, 100 μL of exosome suspension was added to a tube containing 20 μL of CD81 magnetic beads, previously washed with 1 mL of Assay Buffer containing 0.1% bovine serum albumin (BSA, Sigma-Aldrich, Saint Louis, MA, USA) in phosphate-buffered saline (PBS, EuroClone) and incubated at 4 °C overnight with end-over-end mixing. After incubation, the bead-bound exosomes were isolated with the MagnaRack magnetic separator (Thermo Fisher Scientific) and washed twice with 300 μL of Assay Buffer. Isolated CD81-positive exosomes were then labeled with 5 μL mouse anti-human CD63 monoclonal antibody (H5C6)-PE (eBioscience™, Thermo Fisher Scientific) or 20 μL mouse anti-human CD81-PE monoclonal antibody (BD Pharmingen™, BD Biosciences, San Jose, CA, USA). After 45 min incubation at room temperature on an orbital shaker at 1000 rpm, the bead-bound exosomes were washed twice with 300 μL of Assay Buffer, then resuspended in 400 μL of Assay Buffer for flow cytometric detection (Attune™ NxT Acoustic Focusing Cytometer, Life Technologies, Carlsbad, CA, USA). Negative control was performed by staining PBS (vehicle) instead of exosomes. Data collected from the experiments were analyzed using Attune NxT Software version 2.5 (Life Technologies).

A BCA Protein Assay Kit (Thermo Fisher Scientific) was used to measure the concentration of exosomal proteins, according to the manufacturer’s protocol.

### 2.8. Exosomes Labeling and Internalization by Endothelial Cells

Pooled HUVECs (Thermo Fisher Scientific) were cultured according to manufacturer’s instructions using Medium 200PRF completed with Low Serum Growth Supplement Kit (Thermo Fisher Scientific). Exosomes (25 μg/mL) derived from hMSCs grown onto TNs were stained with 10 μM BODIPY™ TR ceramide (Thermo Fisher Scientific) for 20 min at 37 °C. A mix without exosomes was used as the negative control. Excess unincorporated dye was removed from the labeled exosomes with Exosome Spin Columns (MW 3000) (Thermo Fisher Scientific), following the manufacturer’s protocol. The labeled exosomes or control solution were then incubated for 4 h with HUVECs seeded at a density of 5 × 10^4^ cells/cm^2^ in 24-well plates the day before. After incubation, HUVECs were fixed with 4% paraformaldehyde solution in PBS for 10 min, then permeabilized for 10 min in 0.2% Triton X-100 (Sigma-Aldrich) prepared in PBS. After three washing cycles with PBS, the cells were incubated in 2% BSA solution in PBS for 1 h at RT. The cells were then stained with Alexa Fluor™ 488 Phalloidin (Thermo Fisher Scientific) for 1 h at room temperature. Nuclear staining was performed with ProLong™ Glass Antifade Mountant with NucBlue™ Stain (Thermo Fisher Scientific). The cells were observed with the Nikon A1 confocal microscope (Nikon, Minato, Tokyo, Japan) equipped with a 63X objective.

### 2.9. Wound Healing Migration Assay

The migration of HUVECs in vitro was measured using the wound healing assay. Briefly, cells were seeded in 24-well plates at a density of 1 × 10^5^ cells/cm^2^ and cultured in the 5% CO_2_ incubator at 37 °C overnight. Cell monolayers were then scratched using a 200 μL sterile pipette tip. After scratching, each well was washed twice with PBS to remove suspension cells and cultured with serum-free medium. Treatment groups were incubated with 25 μg/mL exo-hMSCs isolated after 7 days of culture onto the different Ti alloy surfaces, whereas the control group was treated with an equal volume of PBS (vehicle). Images of the wound area were taken at 0, 24, and 48 h after the scratch with an EVOS^®^ XL Core Imaging System (Thermo Fisher Scientific). The migration rate was calculated as follows:migration distance (%) = 100% (A_T0_ − A_Th_)/A_T0_(1)
where A_T0_ corresponds to whole wound taken immediately after the scratch and A_Th_ corresponds to whole wound taken after 24 or 48 h [28]. Image J software was used to measure the scratch areas.

### 2.10. RNA Isolation from HUVECs, First-Strand cDNA Synthesis, and Real-Time PCR

Total RNA was isolated from HUVECs treated for 48 h with the different exo-hMSCs preparations using the total RNA Purification Plus kit (Norgen Biotek), as already described in Section 2.4. Real time-PCR was then performed to analyze the expression levels of three genes coding for endothelial-cell-specific markers (Table 2). Data analysis was performed using the 2^ΔΔCt^ method [27] using TFRC as the internal reference. Data are presented as mean fold changes with respect to control samples (HUVECs seeded onto untreated surfaces for 48 h).

### 2.11. Isolation of Total RNA from exo-hMSCs and miScript^®^ miRNA PCR Arrays

Exosomal RNA was isolated from previously purified exo-hMSCs using the Exosomal RNA Isolation Kit (Norgen Bioteck), following the manufacturer’s instructions. Purified total RNA, including miRNAs, was extracted in 20 μL Elution Solution A. Exosomal miRNA expression profiles were analyzed using miScript miRNA PCR Arrays (Qiagen, MIHS-001Z) that investigated the expression of the 84 most abundantly expressed human miRNAs. The cDNA was synthesized from the total RNA using the miScript II Reverse Transcription Kit (Qiagen), according to the user manual. The cDNA was preamplified with the miScript PreAMP PCR Kit, then mixed with QuantiTect SYBR Green PCR Master Mix, miScript Universal Primer, and RNase-free water. The PCR master mix was loaded in 25 μL aliquots across the microarray. The plate was run in a StepOnePlus Real-Time PCR System (Applied Biosystems, Foster City, CA, USA) with the following thermal cycling conditions: initial activation step at 95 °C for 15 min, followed by 15 s at 94 °C for denaturation, 30 s at 55 °C for annealing, and 30 s at 70 °C for extension. The cycle number was set as 40 cycles. The relative expression of the miRNAs was analyzed by the 2^ΔΔCt^ method [27] in comparison to control samples (exo-hMSCs isolated from untreated Ti alloy surfaces after 7 days of culture). Fold changes > 2 or <0.5 and *p*-values < 0.05 were used as the thresholds for selecting differentially expressed miRNAs. To predict miRNA–target gene interactions, the miRNet tool (http://www.mirnet.ca/) was used, whereas pathway enrichment analysis was performed using the Reactome software (http://reactome.org). Here, *p*-values < 0.05 were regarded as statistically significant.

### 2.12. Statistical Analysis

All experiments were conducted in triplicate and repeated three times. Data are expressed as means ± standard deviation (SD). Comparative analysis of two groups was performed using Student’s t-test. Differences between multiple groups were compared using one-way analysis of variance followed by the Bonferroni test. Statistical significance was set at *p* < 0.05.

## 3. Results

### 3.1. Characterizations of Control and Nanostructured Surfaces

SEM images of non-treated (T-untreated) and anodized (TNs) Ti alloys are shown in Figure 1A–D. SEM characterization confirmed the formation of nanotubular features with different diameters. Specifically, 10 V, 20 V, and 30 V applied potentials formed nanotubular structures with diameters of 26.3 ± 3.8, 80.6 ± 4.2, and 138.1 ± 8.7 nm, respectively. The results indicated that when the applied potential increased, so did the nanotubular size on titanium alloy surfaces. In this research, TNs samples with 26.3 ± 3.8, 80.6 ± 4.2, and 138.1 ± 8.7 nm nanotubular diameters were denoted as TNs-25, TNs-80, and TNs-140, respectively, and the control samples were indicated as T-untreated.

AFM scans confirmed that the fabrication of nanostructured surfaces via anodization on Ti alloy altered the nanophase topography of the samples. In AFM images (Figure 1E–H), the 3D nanotubular morphology was evident, specifically for TNs-80 and TNs-140. The root-mean-squared nanophase surface roughness values were measured to be 5.5 ± 1, 17.6 ± 3.2, 28.9 ± 2.4, and 34.5 ± 3.4 nm for T-untreated, TNs-25, TNs-80, and TNs-140, respectively (Table 3). In Figure 1I, SEM images of TN-25, TN-80, and TN-140 surfaces after scratching them to mechanically delaminate the nanotubular oxide layer are reported. The lengths are 345 ± 21 nm, 404 ± 36 nm, and 398 ± 29 nm for TN-25, TN-80, and TN-140, respectively. These results indicate that anodization increased the nanophase surface roughness, and as the nanotubular diameter increased, the anodized surfaces became rougher. However, when root-mean-squared surface roughness values were analyzed to highlight the micrometer-scale topography, no significant differences were observed between anodized and control surfaces. These results signify that the applied anodization process did not alter micrometer scale surface roughness of the surfaces, while increasing the nanophase roughness of the surfaces.

High-resolution XPS spectra of Ti 2p and O 1s for control and anodized surfaces are represented in Figure 2 and Figure 3, respectively. According to the results, the oxide-based surface layer consisted of Ti and niobium oxide with Ti^4+^ and Nb^5+^ states. The Ti^4+^ 2p_3/2_ peaks for untreated, TNs-25, TNs-80, and TNs-140 appeared at 459.0, 458.7, 458.6, and 458.7 eV, respectively. Additionally, the Ti^4+^ 2p_1/2_ peaks for non-anodized, TNs-25, TNs-80, and TNs-140 appeared at 464.7, 464.4, 464.3, and 464.4 eV, respectively. The Nb^5+^ 2p_3/2_ peaks for TNs-25, TNs-80, and TNs-140 appeared at 207.2, 207.1, and 207.2 respectively, while for the non-anodized sample it appeared at 207.7. The results showed that the Ti^4+^ peak intensity increased with an increase in the applied potential. Once the surfaces were sputtered with Ar^+^, peak deconvolution indicated different valence states for Ti underneath the top surface. Peaks appearing around binding energies of 455.9, 456.7, and 458.5 eV were identified and correlated with 2+, 3+, and 4+ valences of Ti to indicate TiO, Ti_2_O_3_, and TiO_2_ species, respectively (Figure 2E–G) [29]. At the same time, the peaks of O 1s at 530.1 and 531.3 eV could be assigned to Ti–O and OH- groups for the nanostructured surfaces (Figure 3). Importantly, the XPS results indicated that after anodization, the intensity of O 1s peaks increased independent of the size of the surface features. This can be explained with the formation of a Ti-based oxide surface layer for the anodized nanotubular surfaces. In addition, the H-bound oxygen (H_2_O) peaks for O 1s appeared at 2.6–2.7 eV stronger binding energies than the lattice oxygen peak, while this peak was not present for the untreated surfaces.

The surface hydrophobicity of the samples after anodization was examined via the water contact angle measurements. The results showed that the water contact angle of the untreated Ti alloy surface was 85.3 ± 7.1°, while TNs-25, TNs-80, and TNs-140 surfaces exhibited water contact angles of 15.8 ± 2.7°, 19.3 ± 3.3°, and 23.5 ± 5.5°, respectively (Table 3). It can be seen that the hydrophilicity of the samples increased after surface modifications. High-resolution XPS analysis of O 1s results (Figure 3) showed that the surfaces consisted of large amounts of oxygen, which could form hydrogen bonds with water and increase hydrophilicity after anodization [30]. Thus, it is clear that the surface nanotopography, surface roughness, and chemical composition of the surfaces altered the water contact angles and thereby affected their hydrophobicity. As a result, the water contact angle of TNs-25 further decreased to 15.8°, indicating the formation of a hydrophilic surface compared to untreated Ti alloy. It should also be noted that these hydrophilic surfaces can potentially inhibit non-specific protein adsorption, platelet adhesion, and activation and can potentially improve blood compatibility [15].

### 3.2. Characterization of hMSCs Grown onto TNs

The morphology of hMSCs was observed by means of SEM, which revealed that the cells adhered and fully covered all the Ti alloy surfaces, forming a complete monolayer in 7 days (Figure 4). However, the cells seeded on unmodified substrates were more flat and showed filopodia that were not particularly pronounced; on the contrary, the cell phenotype became more elongated as the diameter of the nanotubes increased, with the most elongated hMSCs being on TNs-140. These results are in agreement with previous works, showing elongated cells with long filopodia on Ti alloy surfaces with nanotube diameters above 100 nm [31,32,33]. Secretion activity of growth factors is reported on Figure 5 confirming that on the samples cells are able to produce factor related to angiogenesis and osteogenesis.

### 3.3. Characterization of Exosomes Recovered from hMSCs Seeded onto TNs

From a therapeutic point of view, the effects of factors secreted by hMSCs are often inadequate due to their short half-life and rapid degradation rate. Therefore, new strategies aimed at improving the therapeutic efficacy of hMSCs are required. Recently, it has emerged that the beneficial paracrine effects of hMSCs are mediated, at least in part, by exosomes. In this work, exo-hMSCs were isolated from the conditioned media of hMSCs seeded on the different TNs for 7 days through a technology based on Norgen’s proprietary resin, which allows the purification of intact extracellular vesicles. TEM micrographs (Figure 6A) evidenced rounded structures and a similar size range for the negative stained exosomes, with a prevalence of vesicles below 100 nm. Flow cytometry analysis (Figure 6B–E) of exo-hMSCs captured onto anti-CD81 magnetic beads further revealed that these were positive for the tetraspanins CD63 and CD81, as demonstrated by a distinct positive shift of the fluorescence signal beyond the negative controls.

### 3.4. Effects of hMSCs-Derived Exosomes on the Behavior of Endothelial Cells

The hMSCs-derived exosomes were labeled with the red fluorescent dye BODIPY™ TR ceramide, then added into the culture medium of HUVECs. After 4 h of incubation, numerous and granular BODIPY™ TR ceramide-labeled exosomes were observed inside the HUVECs, mostly located in the perinuclear region, suggesting the adsorption and internalization of exosomes by these cells (Figure 7A).

Subsequently, the ability of exo-hMSCs to induce endothelial cell migration in vitro by using the wound healing assay method was studied. Normal HUVECs were plated at confluence; then, after creation of the scratch, cells were stimulated for up to 48 h with exosomes purified from hMSCs grown on the different TNs or with vehicle (Figure 7B). After 24 h, exosomes recovered from hMSCs cultured on the 25 nm, 80 nm, or 140 nm surfaces did not enhance the migration of HUVECs with respect to cells treated with vehicle. On the contrary, the migration rate of endothelial cells was found to be significantly increased at 48 h post-incubation with exo-hMSCs isolated from the TNs as compared to the PBS control. In contrast, the migration distance of HUVECs was not markedly changed by incubation with exosomes recovered from hMSCs cultured on the untreated Ti alloy surfaces at either 24 or 48 h after treatment (Figure 7C). 

### 3.5. miRNA Expression Profiling of Exosomes Recovered from hMSCs Grown onto TNs

In the final part of our work, the focus was on whether exosomes produced by hMSCs grown onto nanomodified Ti alloy surfaces could differentially express a subset of miRNAs when compared to cells seeded onto untreated substrates. Previous studies suggested that hMSCs-derived exosomes may serve as essential mediators of angiogenesis by shuttling genetic materials to recipient cells. Among the constituents of exosomes, miRNAs have a prominent role in regulating gene expression through binding to the 3′UTR of target mRNAs, resulting in translational inhibition or mRNA degradation at the post-transcriptional level [34,35].

Among the 84 miRNAs investigated, and based on the criteria of fold changes >2 or <0.5 and significance at *p* < 0.05, 2 miRNAs were up-regulated and 19 miRNAs were down-regulated in exo-hMSCs isolated from TNs-25, 2 miRNAs were up-regulated and 21 miRNAs were down-regulated in TNs-80 exosomes, and 2 miRNAs were up-regulated and 17 miRNAs were down-regulated in TNs-140 exosomes when compared to untreated Ti alloy surfaces (Table 4). The three different groups shared a signature of 14 differentially expressed miRNAs, as shown in the heat map of Figure 8A.

With the aim of creating an miRNA–target interaction network starting from the 14 differentially expressed miRNAs, the miRNet tool was used, which contains miRNA–gene interaction data collected from miRTarBase v8.0, TarBase v8.0, and miRecords. The network shown in Figure 8B confirms that the potential targets of the differentially expressed miRNAs are genes directly (e.g., VEGFA, bFGF, TGFB1, EGFR) or indirectly (e.g., AKT, HIF1, MTOR, STAT-3) involved in endothelial cell function [36,37,38]. An enrichment analysis was then performed using the Reactome software in order to identify the significant biological pathways for the predicted target genes. A large number of signaling pathways involving several growth factors associated with endothelial cell biology were enriched, such as the PDGF and ERBB2 signaling or the VEGF and FGF receptor signaling pathways (Figure 8C).

## 4. Conclusions

In the present study, we sought to evaluate whether nanotubular structures on Ti alloy surfaces can modulate the secretory activity of hMSCs, which is now recognized as one of the primary mechanisms through which hMSCs exert their therapeutic effects [39]. Our results demonstrated that nanotubes with diameters below 30 nm promoted cell proliferation, whereas dimensions in the 70–100 nm range induced hMSCs differentiation Moreover, we examined the ability of hMSCs seeded onto TNs to express mRNAs of bFGF, VEGFA, HGF, and TGFB1, common angiogenic mediators produced by these cells (Figure 5). Irrespective of the nanostructured surface, the angiogenic regulator bFGF was overexpressed by more than 2-fold in hMSCs grown on Ti alloy compared to cells seeded on TCP for 7 days (Figure 5A). Similarly, the vasculogenic growth factor VEGFA was up-regulated by 3- to 4-fold in cells grown on the Ti alloy surfaces compared to hMSCs seeded in monolayer; the highest expression was measured in cells seeded onto TNs-25 (Figure 5B). In agreement with our results, Gittens and coworkers observed increased production of VEGFA from MSCs cultured over a hydrophilic nanomodified Ti6Al4V surface when compared to a smooth and flat Ti substrate [40]. It has additionally been reported that bFGF enhances endogenous VEGFA production, and that in turn VEGFA is necessary for the bFGF-induced expression of platelet-derived growth factor BB, suggesting cross-talk between the various growth factor signaling pathways [41]. The idea that synergism between several angiogenic molecules is fundamental for a successful angiogenic activity is also supported by Chang and colleagues [42]. In their recent work, the authors demonstrated that coating metal stents with a combination of VEGFA and HGF promotes neovascularization, especially re-endothelialization, after stent implantation in a swine model. HGF is a pleiotrophic factor expressed by cells of mesenchymal origin that induces mitogenesis and survival and prevents apoptosis of endothelial cells through binding to their c-Met receptor [43,44]. Under our experimental conditions, the expression levels of HGF were similar between the investigated Ti alloy surfaces and monolayer cultures (Figure 5C); this possibly indicates that HGF shows stable expression in hMSCs, as proposed by others [45,46,47]. We additionally analyzed the mRNA expression levels of TGFB1 and BMP2. TGFB1, a growth factor that has an important role in cell differentiation and vascular remodeling [48], was up-regulated more than 2-fold in cells grown on all the nanomodified surfaces compared to untreated substrates and monolayer cultures; its expression increased with the nanotube diameter (Figure 5D). Similarly, significant overexpression of BMP2 by 3- to 5-fold was observed in hMSCs grown on Ti alloy surfaces with respect to cells seeded on TCP; the highest expression was measured in cells seeded on TNs-25 and TNs-140 (Figure 5E). BMP2 is a member of the TGFB superfamily of proteins, which is not commonly included among the angiogenic factors secreted by hMSCs. Nonetheless, BMPs and their receptors are associated with vasculogenic and angiogenic processes, since several types of endothelial cells express the BMP2 receptors at high levels [49]. In addition to directly stimulating angiogenesis, BMP2 may have indirect effects on endothelial cells through VEGFA-dependent signaling pathways [50]. Considering that BMP2 trended the same as bFGF and VEGFA, we might assume that both bFGF and BMP2 promote angiogenesis via stimulation of VEGFA. The results described so far suggest that hMSCs adhere robustly on the nanomodified metal surfaces and produce biologically relevant angiogenic factors, providing a first indication that the nanostructuring of the Ti alloy can create a more favorable microenvironment for local angiogenesis. Discussing the results related to the exosomes, we can confirm that both the release and the content of hMSCs-derived exosomes are regulated by the cross-talk between cells and their microenvironment [51]. To test whether the pre-conditioning of hMSCs on nanostructured Ti alloy surfaces may generate exosomes with specific functions, we investigated the effects of exo-hMSCs on the behavior of HUVECs in vitro, since endothelial cells play an important role in the re-endothelialization process of metal stents [52,53]. Among the nanomodified Ti alloy substrates tested in this work, TNs-25 showed the highest endothelial cell migration, followed by TNs-80 and TNs-140. Since migration into the wound site is the first step in re-endothelialization, enhanced HUVEC motility may greatly improve healing after device implantation [53]. Similarly to the observed effects on migration, nanostructured modifications of metal surfaces also modulated the gene expression of endothelial cell-specific markers. In particular, there was a significant increase in VEGFR2 expression following 48 h incubation of HUVECs with exo-hMSCS recovered from TNs-25, TNs-80, and TNs-140 compared to cells treated with exosomes isolated from the unmodified surfaces (Figure 7D). Due to its strong tyrosine kinase activity, VEGFR2 is the main receptor responsible for binding VEGFA to endothelial cells, thereby resulting in the promotion of angiogenesis [11]. The mRNA expression of the junctional molecule CD31 (Figure 7E) and of the secreted glycoprotein VWF (Figure 4F) trended the same as VEGFR2, although it was not statistically significant. Considering the nanotubular Ti substrates examined in this work, TNs-80 exerted greater stimulatory effects on endothelial marker expression than TNs-25 and TN-140, showing in particular significant differences with the larger nanotubular surface. These data are partly in accordance with the study by Beltrán-Partida and colleagues, who reported that anodized Ti6Al4V surfaces with a diameter of around 70 nm improved the angiogenic behavior in terms of the monolayer formation, viability, proliferation, and activation of endothelial factors compared to non-anodized flat counterparts [11]. Nevertheless, in that study, no other nanotubular surfaces were investigated. On the other hand, in a previous study, six different Ti alloy surfaces were produced by anodization with diameters of 15, 20, 30, 50, 70, and 100 nm [54]. The authors demonstrated that Ti surfaces in the nanosize range of 15–30 nm better supported the adhesion, proliferation, mobility, and differentiation of both MSCs and endothelial cells with respect to nanotubular substrates with diameter above 50 nm. The nanostructures with increased hydrophilicity and nanophase roughness can provide a suitable microenvironment for secretion of exosomes and enhance endothelization. Overall, these findings would suggest that the exposure of hMSCs to nanomodified Ti alloy surfaces induces elevated secretion of exosomes, which contain a robust angiogenic signaling profile able to enhance the migration of endothelial cells in vitro and to stimulate the expression of mature endothelial markers.

A panel of miRNAs designated as angiomiRs has been identified to target endothelial cell function, especially angiogenesis [55]. To date, dysregulation of angiomiRs has been mainly investigated in the context of tumor growth and invasion; less is known about the implication of angiomiRs in the physiological process of angiogenesis, such as that occurring following biomaterial implantation [56,57].

Common miRNAs implicated in the MSCs-mediated angiogenesis are those belonging to the let-7 family, which in humans includes 13 members [58]. In this study, we observed a significant down-regulation of some of the let-7 family miRNAs (let-7a-5p, let-7b-5p, let-7c-5p, let-7d-5p, and let-7e-5p) in exo-hMSCs isolated from all the three TNs. In line with our results, the study by Thai and colleagues showed that nanotreatments of TiO_2_ layers result in reduced let-7 miRNAs expression [59]. However, another recent study reported that extracellular vesicles derived from hMSCs contain high levels of these miRNAs, in particular under hypoxic conditions [60]. These contradictory results suggest that let-7s may participate in the control of angiogenesis in multiple ways, which still requires further studies [61].

In our experimental conditions, the pro-angiomiR miR-143-3p was found to be up-regulated in exo-hMSCs recovered from the nanostructured Ti surfaces. In their recent paper, Wang and coworkers demonstrated that miR-143 mediates pro-angiogenic effects in bone marrow endothelial cells by targeting HDAC7, a histone deacetylase involved in the control of endothelial cell growth through β-catenin [62]. In the present study, we additionally identified some miRNAs with recognized anti-angiogenic functions, such as miR-222-3p, miR-15b-5p, and miR-16-5p. MiR-222, together with miR-221, has been shown to inhibit endothelial cell migration, proliferation, and angiogenesis by targeting the stem cell factor receptor c-Kit [63]. MiR-15b-5p and miR-16-5p, which are located in the same cluster on chromosome 3, were down-regulated following hypoxia in human nasopharyngeal carcinoma cells, promoting VEGFA expression, which ultimately facilitated angiogenesis [64]. In the present work, we found that miR-15b-5p and miR-16-5p were down-regulated in hMSC-derived exosomes recovered from TNs-25 and TNs-80, respectively, suggesting that alternate but common mechanisms could be involved in the modulation of angiogenesis by nanomodified surfaces. MiRNA PCR array analysis showed other dysregulated miRNAs that directly target certain positive mediators of angiogenesis. In particular, miR-155-5p and miR-24-3p were found to be significantly down-regulated on all the Ti surfaces, independently of the nanotube dimensions. Down-regulation of miR-155 has been associated with increased CD31 expression with a subsequent improvement in the integrity of endothelial cell junctions [65]. In a recent study, VWF was recognized as a downstream effector of miR-24, at least in osteosarcoma cells [66]. In that experimental model, miR-24 over-expression caused significant down-regulation of VWF at both mRNA and protein levels; conversely, silencing of miR-24 resulted in a significant increase in VWF expression. In the present study, other miRNAs that are not properly defined as angiomiRs, but which have effects on endothelial cells functions, were differentially expressed among the investigated surfaces. In particular, miR-32-5p, mir-125b-5p, miR-146a-5p, and miR-320a were found to be significantly down-regulated on all TNs. It has been shown that miR-32-5p suppresses HUVECs proliferation by acting as a direct regulator of Kruppel-like factor 2, a key molecule in the modulation of endothelial cell proliferation [67]. A similar effect has been proposed for mir-125b-5p, whose under-expression promotes migration and capillary structure formation in HUVECs by regulating the expression of VEGF via erb-b2 receptor tyrosine kinase 2 (ERBB2) [68]. MiR-146a, instead, is well-known for its regulatory role in the immune response and inflammation in monocytes [69], and its down-regulation has been correlated with the inhibition of myocardial cell apoptosis and increased VEGF expression [70]. Regarding miR-320a, its over-expression inhibited the proliferation, migration, and invasion of throphloblast cells and HUVECs while promoting their apoptosis [71].

Collectively, from miRNA PCR array analysis it has emerged that several miRNAs related to angiogenesis are significantly dysregulated on all nanomodified Ti surfaces, irrespective of the nanotube dimensions. On the other hand, some other miRNAs showed a similar trend of up- or down-regulation preferentially on the substrates with the smaller diameters (25 and 80 nm); conversely, few miRNAs were specifically and differentially expressed on the larger substrate. These findings imply that the nanostructured modification of Ti modulates exosome production by hMSCs, and consequently their angiogenic effects, although different mechanisms might be involved. Future studies would be necessary to support the observations that these miRNAs may play a role in the angiogenesis mediated by exo-hMSCs.

Although further studies would be recommended, we believe that these findings may contribute to the design of Ti nanotubular surfaces that favor proper re-endothelialization, which is a key requirement for the long-term success of cardiovascular stent implantation.

## Figures and Tables

**Figure 1 nanomaterials-11-03452-f001:**
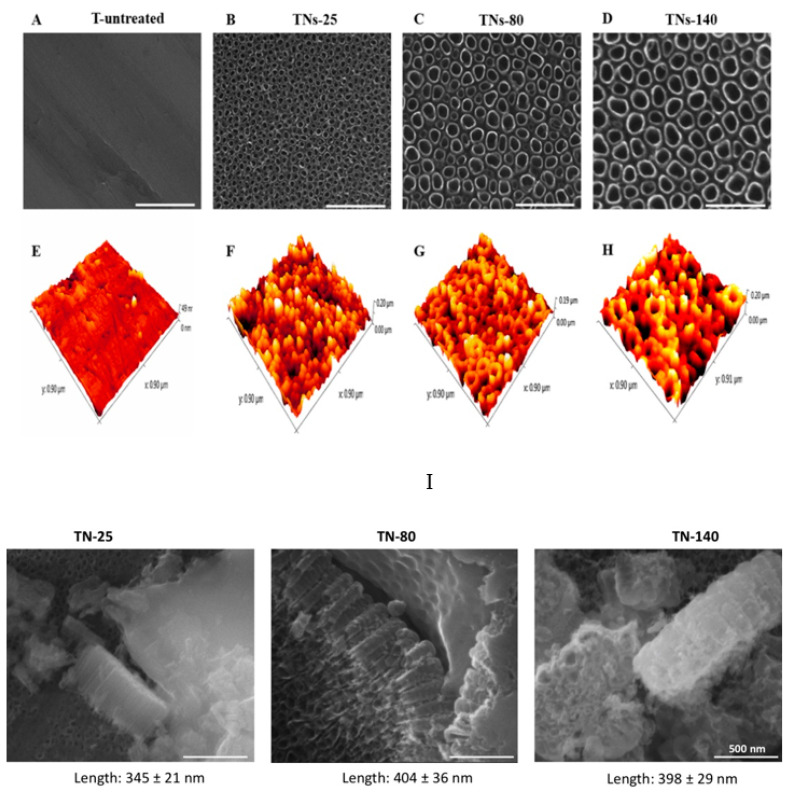
SEM micrographs of (**A**) T-untreated, (**B**) TNs-25, (**C**) TNs-80, and (**D**) TNs-140 surfaces. AFM micrographs of (**E**) T-untreated, (**F**) TNs-25, (**G**) TNs-80, and (**H**) TNs-140 surfaces. (**I**) SEM images of TN-25, TN-80, and TN-140 surfaces after scratching them to mechanically delaminate the nanotubular oxide layer. The lengths are 345 ± 21 nm, 404 ± 36 nm, and 398 ± 29 nm for TN-25, TN-80, and TN-140, respectively. Scale bars are 500 nm.

**Figure 2 nanomaterials-11-03452-f002:**
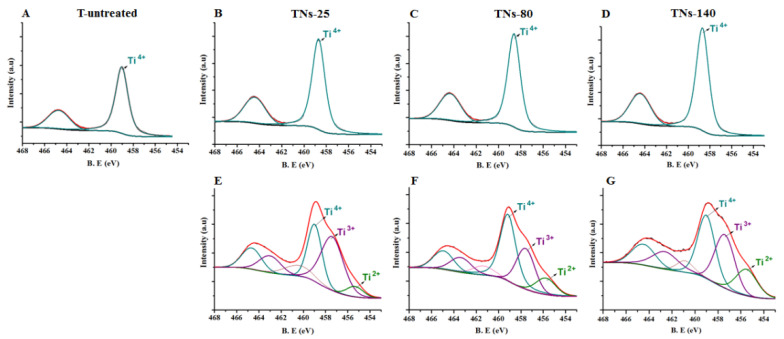
High-resolution XPS spectra of Ti 2p for (**A**) T-untreated, (**B**,**E**) TNs-25, (**C**,**F**) TNs-80, and (**D**,**G**) TNs-140 surfaces (**A**–**D**) prior to and (**E**–**G**) after Ar^+^ sputtering.

**Figure 3 nanomaterials-11-03452-f003:**
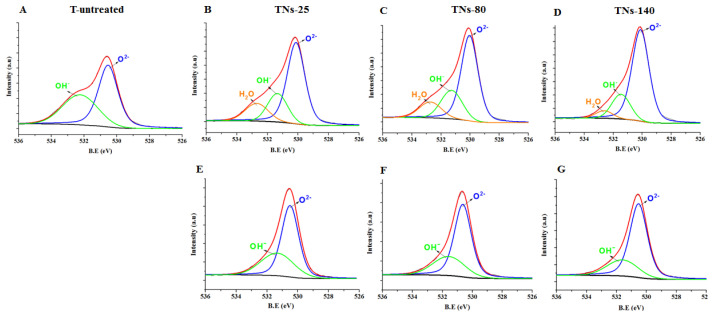
High-resolution XPS spectra of O 1s for (**A**) T-untreated, (**B**,**E**) TNs-25, (**C**,**F**) TNs-80, and (**D**,**G**) TNs-140 surfaces (**A**–**D**) prior to and (**E**–**G**) after Ar^+^ sputtering.

**Figure 4 nanomaterials-11-03452-f004:**
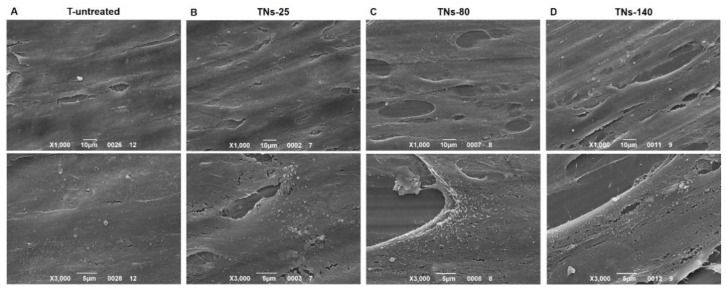
SEM micrographs of the exosome-producing cells. The hMSCs were cultured for 7 days onto (**A**) T-untreated, (**B**) TNs-25, (**C**) TNs-80, and (**D**) TNs-140 surfaces (upper panels ×1000 magnification, lower panels ×3000 magnification).

**Figure 5 nanomaterials-11-03452-f005:**
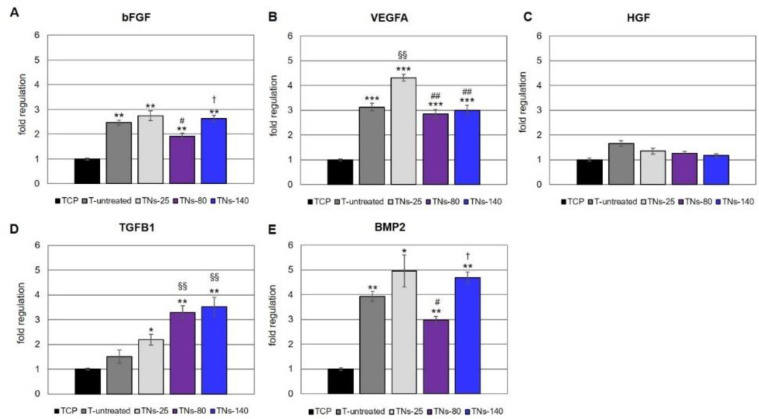
Gene expression profile of angiogenic-related markers in hMSCs seeded onto TNs for 7 days. Expression of (**A**) bFGF, (**B**) VEGFA, (**C**) HGF, (**D**) TGFB1, and (**E**) BMP2 measured by real-time PCR. Data are expressed as means ± SD of the target gene versus reference gene (TFRC) ratio and represented by 2^ΔΔCt^. Note: * *p* < 0.05, ** *p* < 0.01, and *** *p* < 0.001 mark significant changes in gene expression level compared to hMSCs seeded onto TCP for the same culture time; ^§§^ *p* < 0.01 indicates significant changes in gene expression versus hMSCs seeded on T-untreated; ^#^ *p* < 0.05 and ^##^ *p* < 0.01 indicate significant changes in gene expression versus hMSCs seeded on TNs-25; ^†^ *p* < 0.05 indicate significant changes in gene expression versus hMSCs seeded on TNs-80.

**Figure 6 nanomaterials-11-03452-f006:**
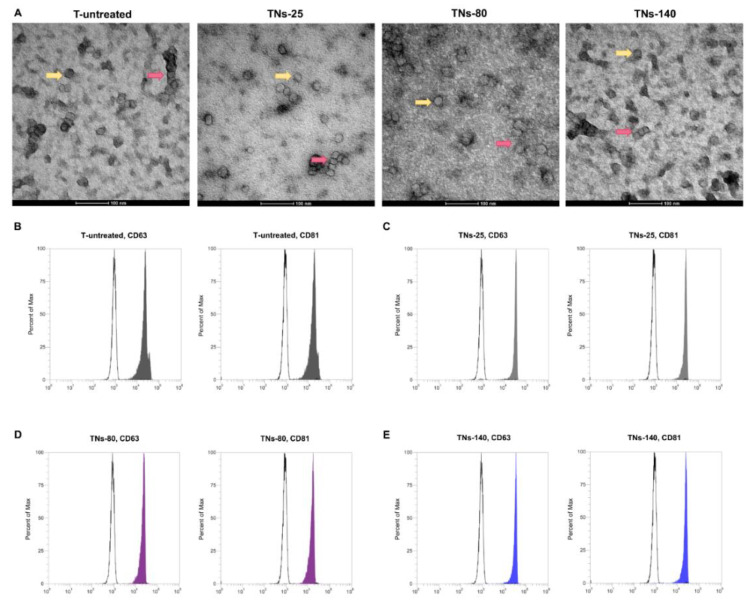
Characterization of exosomes isolated from hMSCs cultured for 7 days onto TNs. (**A**) TEM micrographs of exo-hMSCs recovered from T-untreated, TNs-25, TNs-80, and TNs-140 surfaces. Yellow arrows indicate distinct exosomes, whereas red arrows show exosomes aggregates. Flow cytometry of exo-hMSCs isolated from (**B**) T-untreated, (**C**) TNs-25, (**D**) TNs-80, and (**E**) TNs-140 surfaces showing positivity to CD63 and CD81 exosome surface markers.

**Figure 7 nanomaterials-11-03452-f007:**
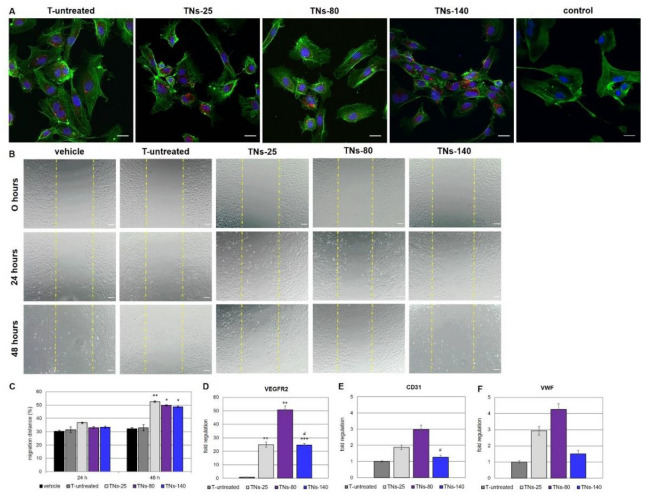
Effects of exo-hMSCs on HUVECs in vitro. (**A**) Internalization of exo-hMSCs by HUVECs. Representative confocal microscope images of HUVECs exposed to BODIPY™ TR ceramide-labeled exosomes (**red**) isolated from hMSCs seeded on T-untreated, TNs-25, TNs-80, and TNs-140 surfaces, or to BODIPY™ TR ceramide-labeled PBS (control). At 4 h post treatment with exosomes, HUVECs were stained with phalloidin (**green**), while nuclei were counterstained with DAPI (**blue**). Scale bar 20 μm. (**B**) Wound healing migration assay of HUVECs treated with the different exo-hMSCs preparations. Representative images of HUVECs migration after treatment for 24 and 48 h with vehicle (PBS), or with exo-hMSCs recovered from T-untreated, TNs-25, TNs-80, and TNs-140 surfaces. Scale bar 100 μm. (**C**) Migration distance calculated at 24 and 48 h post exosomes incubation. Data are expressed as the means ± SD. Note: * *p* < 0.05 and ** *p* < 0.01 mark significant changes in migration distance of HUVECs treated with hMSCs-derived exosomes compared to cells incubated with vehicle for the same culture time. Gene expression profiles of the endothelial cell-specific markers (**D**) VEGFR2, (**E**) CD31, and (**F**) VWF measured by real-time PCR in HUVECs treated for 48 h with the different exo-hMSCs preparations. Data are expressed as means ± SD of the target gene versus reference gene (TFRC) ratio and represented by 2^ΔΔCt^. Note: ** *p* < 0.01, and *** *p* < 0.001 mark significant changes in gene expression level compared to HUVECs incubated with exo-hMSCs isolated from untreated surfaces for the same culture time; # *p* < 0.05 indicates significant changes in gene expression level versus HUVECs incubated with exo-hMSCs isolated from TNs-80.

**Figure 8 nanomaterials-11-03452-f008:**
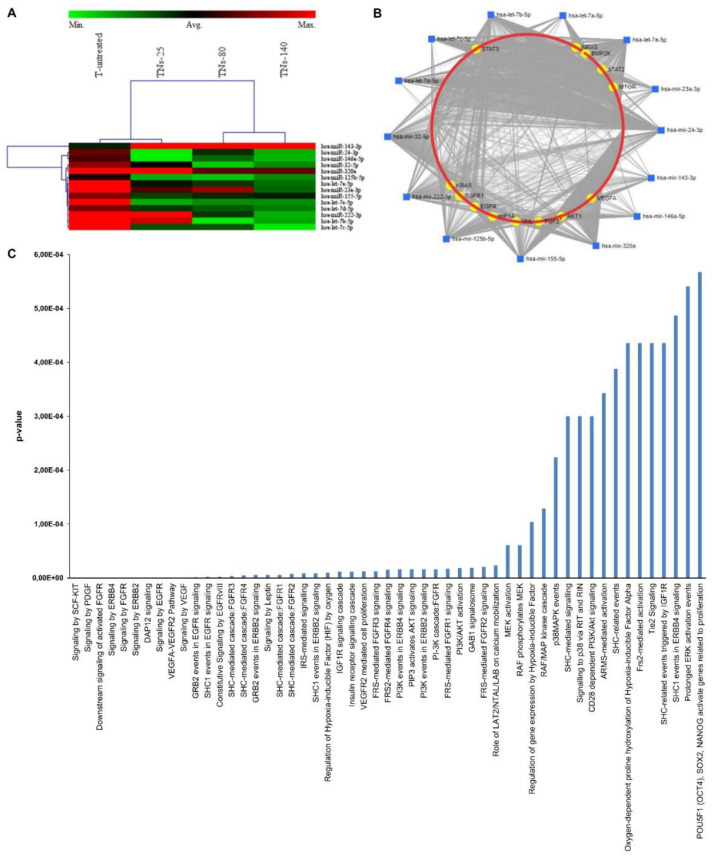
(**A**) Heat map showing the differential expression of miRNAs in TN surfaces versus control group (T-untreated) based on miRNA PCR arrays data. Columns and rows indicate the samples and the specific miRNAs, respectively. Red to green colors indicate the magnitude of gene expression changes. (**B**) The mRNA-miRNA signaling network visualized using miRNet (http://www.mirnet.ca/). Blue squares represent the 14 differentially expressed miRNA, yellow circles indicate the target mRNAs, and their relationship is represented by lines. (**C**) Pathway enrichment analysis using the Reactome software (http://reactome.org). Reactome pathways are displayed on the *x*-axis, and the corresponding *p*-value is shown on the *y*-axis.

**Table 1 nanomaterials-11-03452-t001:** Human-gene-specific primers used in this study.

Gene Name	Sequence FOR (5′-3′)	Sequence REV (5′-3′)
bFGF ^1^	TCCATCCTTTCTCCCTCGTTTC	TCAGTAGATGTTTCCCTCCAATGTTTC
BMP2 ^2^	CCACTAATCATGCCATTGTTCAGAC	CTGTACTAGCGACACCCACAA
HGF ^3^	CTTCAATAGCATGTCAAGTGGAGTG	GCTGCGTCCTTTACCAATGATG
TFRC ^4^	TGTTTGTCATAGGGCAGTTGGAA	ACACCCGAACCAGGAATCTC
TGFB1 ^5^	TGACAGCAGGGATAACACACT	CCGTGGAGCTGAAGCA
VEGFA ^6^	GGACAGAAAGACAGATCACAGGTAC	GCAGGTGAGAGTAAGCGAAGG

^1^ Basic fibroblast growth factor; ^2^ bone morphogenetic protein 2; ^3^ hepatocyte growth factor; ^4^ transferrin receptor; ^5^ transferrin receptor transforming growth factor beta 1; ^6^ vascular endothelial growth factor A.

**Table 2 nanomaterials-11-03452-t002:** Human-gene-specific primers used in this study.

Gene Name	Sequence FOR (5′-3′)	Sequence REV (5′-3′)
CD31 ^1^	TGAGACCAGCCTGATGAAACC	CGATCTCCGCTCACTACAACC
TFRC ^2^	TGTTTGTCATAGGGCAGTTGGAA	ACACCCGAACCAGGAATCTC
VEGFR2 ^3^	GGAGGAGGAGGAAGTATGTGACC	AACCATACCACTGTCCGTCTG
VWF ^4^	ACGTATGGTCTGTGTGGGATC	GACAAGACACTGCTCCTCCA

^1^ Cluster of differentiation 31; ^2^ transferrin receptor; ^3^ vascular endothelial growth factor receptor 2; ^4^ von Willebrand factor.

**Table 3 nanomaterials-11-03452-t003:** Water contact angles, nanophase and micron-phase surface roughness values, and surface areas of T-untreated, TNs-25, TNs-80, and TNs-140 samples.

Sample	Contact Angle (deg °)	Rms (Nanophase Roughness) (nm)	Rms (Micronphase Roughness) (nm)	Surface Area (μm^2^)
T-untreated	85.3 ± 7.1	5.5 ± 1	160 ± 4	0.95 ± 0.07
TNs-25	15.8 ± 2.7 *	17.6 ± 3 *	145 ± 3	1.43 ± 0.08 *
TNs-80	19.3 ± 3.3 *	28.9 ± 2 *	113 ± 2	1.87 ± 0.11 s *
TNs-140	23.5 ± 5.5 *	34.5 ± 3 **	97 ± 4	2.29 ± 0.09 **

Note: * *p* < 0.05 and ** *p* < 0.01 indicate significant differences compared to T-untreated.

**Table 4 nanomaterials-11-03452-t004:** Fold change expression levels and relative *p*-values of miRNAs in exosomes isolated from hMSCs grown on the TNs-25, TNs-80, and TNs-140 surfaces for 7 days versus exo-hMSCs from T-untreated surfaces.

miRNA	TNs-25		TNs-80		TNs-140	
	Fold-Change	*p*-Value	Fold-Change	*p*-Value	Fold-Change	*p*-Value
hsa-miR-142-5p	0.88	0.464	0.85	0.386	1.48	0.112
hsa-miR-9-5p	0.58	0.667	1.01	0.855	1.14	0.641
hsa-miR-150-5p	1.11	0.732	1.84	0.885	1.74	0.616
hsa-miR-27b-3p	1.10	0.770	0.43	0.743	0.44	0.462
hsa-miR-101-3p	1.21	0.850	1.97	0.109	1.32	0.658
hsa-let-7d-5p	**0.40**	**0.002**	**0.33**	**0.016**	**0.25**	**0.000**
hsa-miR-103a-3p	0.20	0.995	0.39	0.959	0.30	0.549
hsa-miR-16-5p	0.98	0.744	0.43	0.517	0.86	0.553
hsa-miR-26a-5p	0.37	0.375	0.40	0.523	0.34	0.231
hsa-miR-32-5p	**0.47**	**0.005**	**0.09**	**0.022**	**0.21**	**0.002**
hsa-miR-26b-5p	0.56	0.672	0.25	0.774	0.27	0.486
hsa-let-7g-5p	0.57	0.720	1.40	0.862	0.61	0.558
hsa-miR-30c-5p	0.36	0.092	0.24	0.180	0.24	0.887
hsa-miR-96-5p	0.96	0.725	0.65	0.678	1.20	0.912
hsa-miR-185-5p	**0.41**	**0.038**	**0.36**	**0.030**	0.44	0.064
hsa-miR-142-3p	0.17	0.592	0.36	0.737	0.72	0.526
hsa-miR-24-3p	**0.04**	**0.011**	**0.45**	**0.020**	**0.17**	**0.000**
hsa-miR-155-5p	**0.37**	**0.003**	**0.42**	**0.011**	**0.44**	**0.002**
hsa-miR-146a-5p	**0.04**	**0.021**	**0.38**	**0.048**	**0.18**	**0.002**
hsa-miR-425-5p	0.57	0.720	0.67	0.843	0.61	0.558
hsa-miR-181b-5p	2.43	0.791	2.43	0.752	3.60	0.522
hsa-miR-302b-3p	0.38	0.681	0.50	0.811	0.38	0.522
hsa-miR-30b-5p	0.60	0.385	0.48	0.473	0.20	0.145
hsa-miR-21-5p	1.26	0.880	0.15	0.651	0.53	0.433
hsa-miR-30e-5p	0.89	0.497	0.83	0.346	0.99	0.797
hsa-miR-200c-3p	0.97	0.728	0.58	0.685	0.63	0.435
hsa-miR-15b-5p	**0.47**	**0.010**	0.85	0.097	0.96	0.545
hsa-miR-223-3p	0.73	0.668	1.72	0.897	0.73	0.526
hsa-miR-194-5p	0.70	0.172	**0.43**	**0.042**	0.60	0.099
hsa-miR-210-3p	0.55	0.393	0.75	0.638	0.84	0.526
hsa-miR-15a-5p	0.57	0.720	0.67	0.843	0.40	0.558
hsa-miR-181a-5p	0.06	0.094	0.27	0.198	**0.04**	**0.032**
hsa-miR-125b-5p	**0.25**	**0.043**	**0.17**	**0.017**	**0.36**	**0.028**
hsa-miR-99a-5p	**0.11**	**0.019**	**0.14**	**0.027**	1.02	0.898
hsa-miR-28-5p	0.57	0.720	0.67	0.843	1.62	0.608
hsa-miR-320a	**0.39**	**0.035**	**0.02**	**0.008**	**0.03**	**0.008**
hsa-miR-125a-5p	**2.43**	**0.012**	0.08	0.290	2.09	0.148
hsa-miR-29b-3p	6.37	0.803	3.98	0.983	0.73	0.564
hsa-miR-29a-3p	1.02	0.917	0.39	0.050	0.48	0.058
hsa-miR-141-3p	0.53	0.298	0.72	0.546	0.54	0.195
hsa-miR-19a-3p	0.83	0.718	1.38	0.895	1.36	0.647
hsa-miR-18a-5p	0.57	0.720	0.67	0.843	1.61	0.608
hsa-miR-374a-5p	0.57	0.720	1.20	0.855	0.75	0.565
hsa-miR-423-5p	0.77	0.251	**0.07**	**0.011**	**0.26**	**0.020**
hsa-let-7a-5p	**0.14**	**0.012**	**0.12**	**0.012**	**0.09**	**0.010**
hsa-miR-124-3p	1.32	0.262	1.35	0.220	1.24	0.412
hsa-miR-92a-3p	**0.34**	**0.010**	**0.45**	**0.049**	0.73	0.064
hsa-miR-23a-3p	**0.15**	**0.013**	**0.18**	**0.014**	**0.05**	**0.009**
hsa-miR-25-3p	0.89	0.476	1.29	0.322	1.16	0.615
hsa-let-7e-5p	**0.08**	**0.013**	**0.11**	**0.017**	**0.11**	**0.011**
hsa-miR-376c-3p	0.90	0.483	**0.09**	**0.025**	0.62	0.085
hsa-miR-126-3p	0.71	0.084	1.21	0.211	0.94	0.618
hsa-miR-144-3p	0.30	0.526	0.49	0.697	0.65	0.469
hsa-miR-424-5p	4.59	0.070	**5.41**	**0.035**	3.19	0.273
hsa-miR-30a-5p	1.10	0.836	0.63	0.115	0.99	0.792
hsa-miR-23b-3p	**0.24**	**0.014**	0.20	0.142	**0.06**	**0.003**
hsa-miR-151a-5p	0.76	0.657	0.16	0.670	0.34	0.402
hsa-miR-195-5p	0.34	0.206	1.04	0.940	0.68	0.268
hsa-miR-143-3p	**4.85**	**0.001**	**4.49**	**0.001**	**16.78**	**0.000**
hsa-miR-30d-5p	0.81	0.210	**0.33**	**0.019**	**0.39**	**0.011**
hsa-miR-191-5p	0.10	0.366	0.49	0.585	0.06	0.222
hsa-let-7i-5p	1.45	0.131	1.11	0.798	1.46	0.123
hsa-miR-302a-3p	0.87	0.352	0.38	0.135	1.33	0.092
hsa-miR-222-3p	**0.20**	**0.016**	**0.03**	**0.008**	**0.01**	**0.008**
hsa-let-7b-5p	**0.19**	**0.015**	**0.01**	**0.009**	**0.03**	**0.008**
hsa-miR-19b-3p	0.77	0.147	0.78	0.158	1.39	0.058
hsa-miR-17-5p	1.20	0.778	2.27	0.846	2.03	0.964
hsa-miR-93-5p	1.61	0.078	1.87	0.064	**2.72**	**0.000**
hsa-miR-186-5p	0.58	0.242	0.96	0.793	1.35	0.373
hsa-miR-196b-5p	0.55	0.545	0.68	0.718	0.75	0.491
hsa-miR-27a-3p	1.72	0.378	1.03	0.388	0.20	0.244
hsa-miR-22-3p	1.80	0.123	0.98	0.751	0.76	0.244
hsa-miR-130a-3p	0.58	0.211	0.74	0.458	0.58	0.188
hsa-let-7c-5p	**0.22**	**0.019**	**0.22**	**0.025**	**0.03**	**0.011**
hsa-miR-29c-3p	0.32	0.430	0.91	0.801	0.98	0.683
hsa-miR-140-3p	0.57	0.084	0.65	0.127	0.87	0.445
hsa-miR-128-3p	0.15	0.565	0.17	0.699	0.11	0.402
hsa-let-7f-5p	0.76	0.716	0.58	0.840	1.14	0.588
hsa-miR-122-5p	8.10	0.583	4.20	0.965	1.10	0.580
hsa-miR-20a-5p	1.04	0.818	0.82	0.701	1.87	0.219
hsa-miR-106b-5p	2.61	0.540	2.26	0.690	1.51	0.855
hsa-miR-7-5p	0.57	0.720	7.99	0.582	2.29	0.656
hsa-miR-100-5p	**0.27**	**0.021**	**0.27**	**0.026**	0.66	0.136
hsa-miR-302c-3p	0.69	0.419	0.97	0.815	0.53	0.215

## Data Availability

Not Applicable.

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
