# Peer review of "Nanostructured Modifications of Titanium Surfaces Improve Vascular Regenerative Properties of Exosomes Derived from Mesenchymal Stem Cells: Preliminary In Vitro Results"

_nanomaterials, 2021, doi:10.3390/nano11123452_

Round 1
Reviewer 1 Report
In the submitted manuscript the authors analyzed three different nanotubular Ti surfaces (TNs), manufactured by electrochemical anodization with diameter of 25, 80, or 140 nm, by seeding the surface with human 24 mesenchymal stem cells. Nanomodifications below 100 nm significantly improved the release of hMSCs exosomes, CD63 and CD81 surface markers.
Authors should try to follow the provided structure in the author guidelines ("Introduction, Materials and Methods, Results, Discussion, Conclusions (optional)"). Currently, the "Discussion" is mixed with the "Results".
Avoid personal pronouns (such as "we") when reporting results.
Try to address limitations, future resulting studies, and possible clinical implications of the results.
Author Response
Authors should try to follow the provided structure in the author guidelines ("Introduction, Materials and Methods, Results, Discussion, Conclusions (optional)"). Currently, the "Discussion" is mixed with the "Results".
Thanks, done
Avoid personal pronouns (such as "we") when reporting results.
Thanks done
Try to address limitations, future resulting studies, and possible clinical implications of the results.
Thanks done

Reviewer 2 Report
The authors Gardin C, Ercan B, Erdogan YK, Zanotti F, De Francesco F, Trentini M, Brunello G, Ferroni L, Zavan B submitted a manuscript entitled “Nanostructured Modifications of Titanium Surfaces Improve Vascular Regenerative Properties of Exosomes Derived from Mesenchymal Stem Cells: preliminary in vitro results”.
The authors presented data, that the modification of the surface of TiO2-alloys by electrochemical anodization led to nanotubular structures which alter the physiological behavior of human mesenchymal stem cells (hMSCs). They show that in consequence the secretion of exosomes by the hMSCs alters. In order to proof the effects of these changes they study the migration of human umbilical vein endothelial cells(HUVEC) as well as the expression of vascular marker genes. Finally, they analyse the miRNA content of the exosomes by an omics-approach with a special focus on angiogenesis-related miRNAs. The authors could demonstrate that the surface modification of the TiO2-alloys leads to an enhanced secretion of exosomes by hMSCs which alter the expression of HUVEC.
The author present an interesting approach which might help to improve endothelialization of stents. Nevertheless, the manuscript has a number of deficiencies that need to be addressed before publication.
I recommend major revision of the work.
Concerns
- The authors present SEM and AFM pictures to demonstrate the specific changes in surface morphology of the TiO2-alloys after electrochemical anodization. Is the depths of the nanotubules different with regard to the treatment?
- The authors present the primer sequences for expression analysis of hMSCs in table 1 and for expression analysis of HUVEC in table 2. The annotation in the legends does not fit to the gene names in the table. In addition the primer sequences are wrong. E.g. table 2: CD31: the presented sequences are FGF2 sequences (NCBI blast). A thorough review of the sequences is urgently needed!
- The authors presented data on the expression of angiogenesis-related genes, but they only present statistical analysis on untreated TiO2 versus treated TiO2 (Figure 5). Are there any significant differences between the 3 treatment groups?
- Is one of the treatment groups preferable?
- In figure 6 the authors claimed that nanoparticles and exosomes are visible on TEM micrographs. They marked the structures by yellow arrows (nanoparticles) and red arrows (exosome aggregates).To be honest, I could not see a different appearance.
- The authors claim that the different exosome preparations affect the migration properties of HUVEC and present micrographs in figure 7B. It is impossible to identify any cells. There are only 5 greyish areas.
- In most of the pictures the lettering is very hard to read (Figure 4, Figure 7C and 7D, Figure 8B)
- The manuscript should be revised in terms of superscript and subscript
Author Response
- The authors present SEM and AFM pictures to demonstrate the specific changes in surface morphology of the TiO2-alloys after electrochemical anodization. Is the depths of the nanotubules different with regard to the treatment?
Thank you for bringing it up. During the anodization process, the applied voltage and anodization duration affected nanotubular length. Nanotubular lenghts are 345 ± 21 nm, 404 ± 36 nm and 398 ± 29 nm for TN-25, TN-80 and TN-140, respectively. This was indicated in Fig. 1 I. In this study, our aim was to reveal the effect of nanophase topography and nanotubular size on exosome release. Our results showed that fabrication of nanotubular features on titanium alloy surfaces changed its hydrophilicity and roughness, which enhanced cellular spreading, proliferation and exosome secretion. Though altering anodization parameters also resulted in changes in the nanotubular lengths, assessing the influence of nanotube length on exesome secretion is outside the scope of this manuscript. Having this said, we are currently working on another manuscript to control electrochemical parameters to systematically alter nanotube diameters while keeping nanotube lengths constant.
- The authors present the primer sequences for expression analysis of hMSCs in table 1 and for expression analysis of HUVEC in table 2. The annotation in the legends does not fit to the gene names in the table. In addition the primer sequences are wrong. E.g. table 2: CD31: the presented sequences are FGF2 sequences (NCBI blast). A thorough review of the sequences is urgently needed!
We have corrected the annotations in the legends. We have checked all reported sequences and corrected the wrong ones (see Table 1 e Table2)
- The authors presented data on the expression of angiogenesis-related genes, but they only present statistical analysis on untreated TiO2 versus treated TiO2 (Figure 5). Are there any significant differences between the 3 treatment groups?
Is one of the treatment groups preferable?
In figure 5 the present statistical analysis refers to the comparison between nanomodified surfaces and tissue culture polystyrene (TCP), and the comparison between TNs-25 and the other nanomodified surfaces. We have added the comparison between nanomodified surfaces and untreated surface, and between TNs-80 and TNs-140.
Cells affected by nanomodified surfaces exhibit a different expression profile from those grown on untreated substrate or as monolayer culture. It appears that nanotubular structures having 25 nm diameter are sufficient to trigger change in the expression of angiogenesis-related genes.
- In figure 6 the authors claimed that nanoparticles and exosomes are visible on TEM micrographs. They marked the structures by yellow arrows (nanoparticles) and red arrows (exosome aggregates).To be honest, I could not see a different appearance.
- Yellow arrows indicate single exosomes, while red arrows indicate clusters of exosomes. Both arrows indicate exosomes, the only difference being the number of exosomes: single (yellow arrows) or aggregate (red arrows).To clarify, we replaced “nanoparticles” with “exosomes” in the caption of Figure 6.
- The authors claim that the different exosome preparations affect the migration properties of HUVEC and present micrographs in figure 7B. It is impossible to identify any cells. There are only 5 greyish areas.
We have improved contrast and brightness in images of the migration assay to better identify cells.
- In most of the pictures the lettering is very hard to read (Figure 4, Figure 7C and 7D, Figure 8B)
We have enlarged the lettering Figure 4, Figure 7C and 7D and enhanced dpi of Figure 8
- The manuscript should be revised in terms of superscript and subscript
Thanks done.

Reviewer 3 Report
The study aimed to evaluate the effect of Ti surface modification with different size of nanotubes to improve the endothelization of cardiovascular stents and their clinical outcome. From this view, the rationale of the study was not sufficiently justified. Why the Authors did select MSCs? What is the role of MSCs during stent endothelization? Why the Authors did determine the expression of osteogenesis-related factors? Their connection to the endothelization is very questionable and should be explained.
Line 28. Expressed significantly higher levels of angiogenic and osteogenic …..
Line 135. MSCs instead “stem cells”
Lines 274-276. The authors must approve the normal distribution before the application of t-test and ANOVA.
Author Response
The study aimed to evaluate the effect of Ti surface modification with different size of nanotubes to improve the endothelization of cardiovascular stents and their clinical outcome. From this view, the rationale of the study was not sufficiently justified. Why the Authors did select MSCs? What is the role of MSCs during stent endothelization? Why the Authors did determine the expression of osteogenesis-related factors? Their connection to the endothelization is very questionable and should be explained.
Aim of the present work was not the evaluation of the endotheliazation of stent Ti surfaces but to evaluate if nanostructures of Titanium based scaffolds could affects the vascular properties of the exosomes delivered from MSC. Since MSC are the first cells involved on tissue regeneration we used them to evaluate the ability of the nano treatment cto induce a positive effect on vascularization of the scaffolds. The ability of MSC to rich in vitro an osteogenic phenotype in one of the key request to defined them as Mesenchymal stem cells.
Line 135. MSCs instead “stem cells”

Round 2
Reviewer 1 Report
All points raised have been implemented by the authors.
Reviewer 2 Report
The authors have addressed most of the reviewer's comments. In table 1 the term "transferrin receptor" in front of "transforming growth factor beta" has to be cleared. After this correction the manuscript can be published as is.